# Interaction of Polyphenylsilsesquioxane with Various β-Diketonate Complexes of Titanium by Mechanochemical Activation

Vitalii Libanov [1,*] , Alevtina Kapustina [1], Nikolay Shapkin [1], Anna Tarabanova [1] and Anna Rumina [2]

[1] Institute of Science-Intensive Technologies and Advanced Materials, Far Eastern Federal University, 690950 Vladivostok, Russia; kapustina.aa@dvfu.ru (A.K.); shapkin.np@dvfu.ru (N.S.); tarabanova.ae@dvfu.ru (A.T.)

[2] Laboratory of Hydrochemistry, V.I. Il'ichev Pacific Oceanological Institute, Far Eastern Branch Russian Academy of Sciences, 690041 Vladivostok, Russia; ryumina.aa@poi.dvo.ru

* Correspondence: libanov.vv@dvfu.ru; Tel.: +7-902-480-47-83

**Abstract:** In the present work, we studied the interaction of polyphenylsilsesquioxane with various β-diketonate complexes of titanium by mechanochemical activation. Polyphenylsilsesquioxane, bis-(2,4-pentanedionate) titanium dichloride, bis-(1-phenyl-1,3-butanedionate) titanium dichloride, and bis-(1,3-diphenyl-1,3-propanedionate) titanium dichloride were used as starting reagents. Various chemical and physicochemical methods of analysis were used to study the synthesis products. The composition of the obtained compounds has been determined. It is shown that under conditions of mechanochemical activation, high-molecular-weight products with a Si/Ti ratio different from the specified ones are formed. In addition, under the action of mechanical stresses, the initial titanium complexes (except for acetylacetonate complex) polymerize with the formation of coordination of high-molecular-weight compounds, which are destroyed by the addition of ethyl alcohol. It has been established that with an increase in the volume of the organic ligand, titanium atoms enter the polymer siloxane chain to a lesser extent. This work is aimed at finding efficient and environmentally friendly methods for the synthesis and modification of organometallic macromolecular compounds.

**Keywords:** titaniumphenylsiloxanes; mechanochemical synthesis; green chemistry; polyphenylsilsesquioxane; titanium dichloride b-diketonates

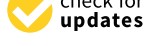



## 1. Introduction

The method of mechanochemical activation is becoming more and more popular not only in the synthesis of inorganic and organic compounds [1–8], but also in the modification of element organic polymers [9–14]. This method has been widely used as one of the methods for modifying macromolecular compounds, as well as for processing secondary polyolefins [15] and for removing persistent organic pollutants [16].

Titansiloxanes are of considerable scientific and practical interest. They have been widely used as Ziegler–Natta catalysts [17–20], oxidation and polymerization catalysts [21–23], and for epoxidation [24,25]. In addition, they are precursors of materials for medicine and microbiology [26,27].

The authors of [28] used diethoxydimethylsilane and titanium isopropoxide to obtain polymer nanocomposites by the sol-gel process. The authors found that, depending on the Ti/Si molar ratio, gels and films (both flexible and brittle) with an average thickness of 20 μm are obtained, which can subsequently be used for laser matrices. The structural model of the gel proposed by the authors is a siloxane chain as the basis of a matrix framing titanium dioxide particles. The addition of coumarin 4 and rhodamine 6G to such films gives them optical properties. Titanium dioxide nanoparticles, derivatized by trifunctional siloxanes, exhibit hydrophobic properties (contact angle of about 140–150°), full buoyancy above water, and excellent self-cleaning properties [29].

For the first time, the authors of [30] reported the synthesis of high-molecular-weight ti-titansiloxanes. Andrianov and co-workers found that the co-hydrolysis of dimethyldichlorosilane, diethyldichlorosilane, and methylphenyldichlorosilane with bis-(acetylacetonate) titanium dichloride in the absence of an acceptor proceeds with the formation of polymers with titanium siloxane bonds. The authors also showed that 60% of the initial titanium complex polymerizes independently and does not interact with silanes. When carrying out the reaction with pyridine, the yield of titansiloxanes increases to 70%.

There are also several difficulties in the synthesis of titanesiloxanes. Thus, the authors of [31] showed that if the silicon atom in the initial organosilicon compound has a tert-butyl group, then a derivative with one or two cyclopentadienyl rings at the metal atom can be obtained. This cannot be performed with ethyl or phenyl substituent on the silicon atom.

The presence of a branched organic group in the silicon atom makes it possible to use, in these reactions, not only diols but also organosilanetriols [32]. The fluorenyl group has the greatest screening effect, the use of which makes it possible to exclude the self-condensation of silanetriols during their interaction with titanium alkoxides, leading to the formation of framework titanosiloxanes [33,34].

Titanesiloxanes can also be obtained by mixing titanium dioxide nanoparticles with a readymade polymer matrix, as performed by the authors of [35–38], by sol-gel synthesis based on titanium alkoxides [39–41]. The processes of direct interaction of titanium alkoxides with hydromethylsiloxane, as well as cleavage of the siloxane bond in the copolymer hydromethylsiloxane-phenylmethylsiloxane under the action of tetrabutoxytitanium, have been studied [42].

Titanesiloxanes are formed by the interaction of organotitanium oxides with silanols of various functionalities [43]. Here, the use of tri-functional tert-butylsilanetriol as a starting organosilicon compound at room temperature leads to the formation of intermediate complexes, which, upon heating, form compounds with an adamantane structure. Gunji [44] synthesized cyclic titanium siloxanes based on organosilicon diols and 2,4-pentanedionate titanium complexes. Pyrolysis of such compounds led to the formation of ceramics comprising amorphous silicon dioxide and crystalline titanium dioxide.

"Classical" reactions of tri-functional organosilicon compounds with tri-functional organotitanium compounds should lead to the formation of amorphous ladder polymers. However, the use of compounds with bulky organic groups (triphenylmethylsilanthriol and cyclopentadienyltitanium trichloride) in [45] led to the formation of framework eighteen-membered cyclic compounds. The condensation reactions of silanetriols and titanium alkoxides were also studied in [33,34,46,47]. The authors found that the yields and the composition of the products depend on the synthesis temperature, the type and the nature of the solvents, and the nature of the organic substituent at the silicon and titanium atoms.

For the synthesis of titanesiloxanes, the so-called exchange decomposition reactions leading to the formation of crystalline low-molecular-weight products were used [48].

In this work, for the synthesis of titanesiloxanes, a new method based on the mechanochemical interaction of the starting compounds was applied.

## 2. Materials and Methods

In this work, we used commercial solvents, which were purified according to standard methods. The physical constants coincided with data from the literature. As starting reagents, we used chemically pure titanium tetrachloride (Eko-tec, Moscow, Russia), pentane-2,4-dione (Ekos-1, Moscow region, Staraya Kupavna, Russia), 1-phenylbutane-1,3-dione (Alfa Aesar, Germany), and 1,3-diphenylpropane-1,3-dione (Alfa Aesar, Kandel, Germany). We distilled phenyltrichlorosilane at a temperature of 201–202 °C.

### 2.1. Synthesis of Starting Reagents

Synthesis of polyphenylsilsesquioxane. Polyphenylsilsesquioxane (PPSSO) was synthesized by the procedure in [8]. Yield, 94.3%. Found, %: C 53.8; Si 20.8. $[C_6H_5SiO_{1.5}\cdot0.26H_2O]_n$. Calculated, %: C 53.8; Si 20.9.

Synthesis of dichloro-bis-(4-oxopent-2-en-2-olate) titanium (IV) (bis-(acetylacetonato) titanium dichloride). First, 0.1 mol of titanium tetrachloride was slowly added to a mixture comprising 50 mL of chloroform, 50 mL of carbon tetrachloride, and a small excess of acetylacetone (from the calculated value). The synthesis was carried out with continuous stirring for two hours. The resulting red precipitate was filtered and dried in a vacuum oven at a temperature of 75 °C. The melting temperature was 185 °C (with decomposition) [49]. Found/calculated, %: C 37.9/37.9, Ti 15.1/15.1, Cl 22.7/22.4. $^1$H NMR (CDCl$_3$, δ): 2.18 (s, 12H, -CH3), 6.06 (s, 2H, γ-H).

Synthesis of dichloro-bis-(3-oxo-1-phenylbut-1-en-1-olate) titanium (IV) (bis-(benzoylacetonato) titanium dichloride). First, 0.1 mol of titanium tetrachloride was slowly added to a solution of 0.2 mol of 1-phenylbutane-1,3-dione in 50 mL of carbon tetrachloride. Synthesis was carried out with continuous stirring for two hours. The resulting red precipitate was filtered and dried in a vacuum oven at a temperature of 75 °C. The melting point was 209–210 °C (with decomposition) [50]. Found/calculated, %: C 54.5/54.4, Ti 10.8/10.8, Cl 16.3/16.0. $^1$H NMR (CDCl$_3$, δ): 2.21 (s, 6H, -CH3), 6.66 (s, 2H, γ-H), 7.39–7.64 (m, 4H, ar), 7.76–7.94 (m, 2H, ar), 7.98 (m, 4H, ar).

Synthesis of dichloro-bis-(3-oxo-1,3-diphenylprop-1-en-1-olate) titanium (IV) (bis-(dibenzoylmethanato) titanium dichloride). First, 0.1 mol of titanium tetrachloride was slowly added to a solution of 0.2 mol of 1,3-diphenylpropane-1,3-dione in 50 mL of carbon tetrachloride. Synthesis was carried out with continuous stirring for two hours. The resulting red precipitate was filtered and dried in a vacuum oven at a temperature of 75 °C. The melting point was 262–264 °C [50]. Found/calculated, %: C 64.0/63.7, Ti 8.5/8.5, Cl 12.9/12.5. $^1$H NMR (CDCl$_3$, δ): 6.88 (s, 2H, γ-H), 7.32 (m, 6H, ar), 7.48 (m, 8H, ar), 7.57 (m, 2H, ar), 8.01 (m, 4H, ar).

### 2.2. Reaction of Polyphenylsilsesquioxane with Titanium Dichloride β-Diketonates

All syntheses were carried out in a Pulverisette 6 planetary monomill (FRITSCH, Idar-Oberstein, Germany). The activating packing was stainless steel balls that were 0.8 cm in diameter and 4.05 g in weight. The packing weight to payload ratio was approximately 1.8–2. Mechanochemical activation was carried out at a frequency of 600 rpm (10 Hz) for three minutes. Each of the syntheses was carried out in two parallel syntheses; the results of each parallel synthesis are reproducible with an accuracy of 0.5% (determined by product yields and results of chemical analysis).

Interaction of polyphenylsilsesquioxane with dichlorobis-(4-oxopent-2-en-2-olate) titanium (IV). We charged the reactor of the planetary mill with 0.025 mol of polyphenylsilsesquioxane and the same amount of complex 1 (synthesis 1). The initial Si/Ti ratio was 1:1. After activation, the reaction mixture was divided into soluble and insoluble fractions by extraction with toluene in a Soxhlet apparatus. The solvent was distilled off from the soluble fraction. Both fractions were dried in a vacuum oven at a temperature of 75 °C. The analysis of the obtained compounds is given in the section Results.

Interaction of polyphenylsilsesquioxane with dichlorobis-(3-oxo-1-phenylbut-1-en-1-olate) titanium (IV). We charged the reactor of the planetary mill with 0.01 mol of polyphenylsilsesquioxane and the same amount of complex 2 (synthesis 2). The initial Si/Ti ratio was 1:1. After activation, the reaction mixture was transferred into a beaker with 100 mL of carbon tetrachloride. The solution was evaporated on a rotary evaporator and dried in a vacuum oven at a temperature of 75 °C until constant weight (fraction 1). The insoluble precipitate was additionally treated with chloroform and filtered. After distilling off the solvent and drying in a vacuum oven at a temperature of 75 °C, a soluble fraction (fraction 2) and a fraction insoluble in chloroform (fraction 3) were isolated. The analysis of the obtained compounds is given in the section Results.

Interaction of polyphenylsilsesquioxane with dichlorobis-(3-oxo-1,3-diphenylprop-1-en-1-olate) titanium (IV). We charged the reactor of the planetary mill with 0.01 mol of polyphenylsilsesquioxane and the same amount of complex 3 (synthesis 3). The initial Si/Ti

ratio was 1:1. The division of fractions was carried out similarly to the previous synthesis. The analysis of the obtained compounds is given in the section Results.

### 2.3. Characterization

Elemental analysis of the obtained compounds

The gravimetric method was used to carry out the determination of silicon [51]. The quantitative determination of titanium was carried out using the method of back titration [52] after the mineralization of polymers with a mixture of concentrated nitric and perchloric acids. Chlorine was determined according to the Schoniger method [53]. Elemental analysis for carbon was carried out on a Flash EA 1112CHN/MAS200 carbon, hydrogen, and nitrogen analyzer (ThermoFinnigan MAT GmbH, San Jose, CA, USA).

Gel permeation chromatography

Gel permeation chromatography (GPC) was performed on a 980 mm long column with a diameter of 12 mm filled with a copolymer of polystyrene and 4% divinylbenzene. The diameter of the grains was 0.08–1 mm. The eluent was toluene and the flow rate was 1 mL/min. The size of the sample was ~0.2 g. The detection was carried out by a gravimetric method according to the content of dry residue in the fractions. A portion of the substance was dissolved in 2 mL of toluene and passed through a column. Fractions of the solution were collected in 3 mL and the solvent was removed in a drying cabinet. The column was preliminarily calibrated with substances with different molecular masses: polydimethylsiloxane $H[Me_2SiO]_{30}OH$ (M = 2238), octaphenylcyclotetrasiloxane $[Ph_2SiO]_4$ (M = 792), hexaphenylcyclotrisiloxane $[Ph_2SiO]_3$ (M = 594), and benzoic acid (M = 122).

IR spectroscopy

IR spectra were recorded on a Spectrum BX 400 FT-IR spectrometer (Perkin Elmer, Shelton, CT, USA) in potassium bromide.

X-ray phase analysis

X-ray phase analysis was performed on a MiniFlex II X-ray diffractometer (RIGAKU, Tokyo, Japan). X-ray tube—Cu, power 0.45 kW, generator power 1 kW, goniometer geometry—vertical, radius—150 mm, scanning step (2θ) 0.01.

Nuclear magnetic resonance

NMR spectra were recorded on an Avance 400 MHz high-resolution spectrometer (Bruker, Bremen, Germany) on $^1H$ and $^{13}C$ nuclei at various operating frequencies. Deuterated chloroform and dimethyl sulfoxide-$d_6$ were used as solvents.

Scanning electron microscope (SEM)

Electron microscopy was carried out using a Zeiss EVO 60 scanning electron microscope (Zeiss, Oberkochen, Germany) at different ranges of accelerating voltages.

The product yield per element was determined by the degree of entry of the element into the polymer chain. The yield was determined by the formula:

$$Yield, \% = \frac{m_f W}{m_i},$$

where $m_i$ is the mass of the element introduced into the reaction; $m_f$ is the mass of the fraction; and $W$ is the found content of the element in percentage.

### 3. Results

Introducing titanium atoms into the siloxane chain is a difficult task because titanium compounds have several properties that lead to side reactions, such as isomerization of the starting components of the reaction, as well as reactions between the starting materials and the solvent. Mechanochemical activation ensures the elimination of solvents both at the stage of synthesis and (sometimes) at the stage of isolation of reaction products. In our work, we used titanium derivatives in which bulky β-diketonate groups surrounded the atom introduced into the siloxane chain. It was assumed that the reaction would proceed according to the radical ion mechanism described in [9,10], and that the polymerization of

the initial titanium complexes, described in [30], would be absent due to the exclusion of the solvent in the process of mechanochemical activation.

### 3.1. Synthesis Based on Dichloro-bis-(4-oxopent-2-en-2-olate) Titanium (IV)

When isolating the products of synthesis 1 based on PPSSO and complex 1, two fractions were obtained. The toluene-soluble fraction (fraction 1.1) was a bright orange powder, while the insoluble fraction (fraction 1.2) was a beige powder. The elemental composition of the fractions is shown in Table 1.

**Table 1.** Data from elemental analysis of products of synthesis 1.

| Fraction | W, % | Found/Calculated, % | | | | | Yield, % | |
|---|---|---|---|---|---|---|---|---|
| | | **Ti** | **Si** | **C** | **Cl** | **Si/Ti** | **Ti** | **Si** |
| 1.1 | 31.60 | $(L^1{}_2TiO)(PhSiO_{1.5})_9(TiO_2)_2(PhSiCl(O))(PhSiO_{0.5}(OH)Cl)$ | | | | | | |
| | | 7.6/7.5 | 16.5/16.2 | 47.6/47.9 | 3.7/3.7 | 3.6 | 18.54 | 67.58 |
| 1.2 | 68.40 | $[(L^1{}_2TiCl_2)_5(TiO_2)_{0.28}]\cdot[(PhSiO_{1.5})(SiO_2)_{0.18}]$ | | | | | | |
| | | 14.9/14.5 | 1.7/1.9 | 38.4/38.4 | 20.3/20.3 | 1:5.1 | 78.83 | 15.37 |

where L$^1$—

As seen from the data given in Table 1, the obtained Si/Ti ratio in compound 1.1 differed from the specified one and amounted to Si/Ti = 3.6. In the IR spectrum (see Figure S1 in Supplementary Materials), there is a broad absorption band near 1000–1028 cm$^{-1}$, corresponding to vibrations of the siloxane bond. Some shift of the vibrations of the siloxane bond towards lower frequencies show stressed cycles in the macromolecule. The broad absorption band at 1132 cm$^{-1}$ corresponds to the vibrations of the Si-C$^{Ph}$ bond. The presence of the Ti-O-C bond increases the absorption in the region of 1000–1130 cm$^{-1}$.

Low-intensity vibrations of the bonds of acetylacetonate groups are manifested near 1357 cm$^{-1}$ (-C-O-Ti) and 1527 cm$^{-1}$ (C=O). Stretching and bending vibrations of CH bonds in the acetylacetonate fragment and phenyl substituent are manifested in the regions of 1430, 3074, 3051, and 3008 cm$^{-1}$; vibrations of the C=C bond in the acetylacetonate fragment are overlapped by the same bonds of the phenyl substituent and are manifested in the 1595 cm$^{-1}$ region. The broad absorption band at 3431 cm$^{-1}$ corresponds to free vibrations of the hydroxyl group bound to the silicon atom [54]. Noteworthy is the absence of an absorption band near 540 cm$^{-1}$, which is characteristic of symmetric vibrations of the C–O–Ti bond [55]. This is also confirmed by the data from the $^1$H NMR spectroscopy. Vibrations in the 775 cm$^{-1}$ region correspond to vibrations of the Ti-O bond in an octahedral environment.

According to the data from the $^1$H NMR spectroscopy, the following signals are present in this compound: a multiplet between 7.13–7.41 ppm. (55 H), corresponding to the chemical shifts of protons in the aromatic ring, two singlets at 5.95 and 5.77 ppm. (2H), corresponding to the chemical shift of protons in the γ-position of the acetylacetonate ring, and singlets at 2.03 and 2.09 ppm. (12H), corresponding to the chemical shifts of the protons of the methyl group of the acetylacetonate ring.

According to the data from the gel permeation chromatography, the molecular weight of the obtained compound is about 2000. Whereas compound 1.1 contained reactive functional groups (Si-OH, Si-Cl, and Ti-L$^1$), we attempted to heat this compound. Upon heating for 20 min at 200 °C, the absolute weight of the polymer decreased by 8.5%, and the molecular weight increased to 6000. This shows condensation processes occurring at elevated temperatures, including those associated with the presence of hydrolytically unstable Si-Cl bonds. In the IR spectra of the polymer obtained upon heating, there were no absorption bands characteristic of the vibrations of the Si-OH and Si-Cl bonds, but acetylacetonate groups were kept. The vibrations of the bonds of the ligand fragment

disappeared when the polymer was heated to 300 °C, after which the polymer ceased to lose its absolute weight. Thus, the following formula can correspond to compound 1.1 (Scheme 1):

**Scheme 1.** Structural units of compound 1.1.

We previously showed that the structure of polymetallic organosiloxanes is like the structure of layered silicates, including examples in [56]. The supramolecular structure of such polymers contains ordered regions (lamellas) in which the chains of macromolecules are folded like a ribbon. Parameters, such as the size of the coherent scattering region (CSR) determined by X-ray diffraction analysis, can be used to study the supramolecular structure. The CSR corresponds to the size of the lamella in a certain direction, and the first and second reflections on the X-ray diffraction patterns give us the dimensions of the lamella.

It is known [57] that the diffraction pattern of polyphenylsilsesquioxane contains two diffraction maxima. The first maximum characterizes the interchain distances in the equatorial plane perpendicular to the axes of the macromolecule. The second maximum characterizes mainly intrachain distances of different orientations. In the same work, the authors give an equation for calculating the average diameter of macromolecules (l) according to the position of the first maximum on the intensity distribution curves (the first amorphous halo in the diffractogram):

$$\sqrt{3}lsin\theta = \lambda \tag{1}$$

Besides the average diameter of macromolecules, we calculated the size of the coherent scattering region (L) using the Scherrer formula [58], and the cross-sectional area of the macromolecule was calculated using the Miller–Boyer equation [59]. It was shown in [60] that polyphenylsiloxanes belong to type B according to the Boyer–Miller classification, which corresponds to the coefficients $k_1 = 0.06$ and $k_2 = 0.61$ in the equation $log(d) = k_2 lg(s) + k_1$.

According to the X-ray phase analysis data (Table 2), compound 1.1 is amorphous.

**Table 2.** Data from X-ray phase analysis of PPSSO and fraction 1.1.

| Polymer | Diffraction Peak | | $2\theta(d_1)$, Degree | $l$, nm | $D_{CSR}$, nm | $S$, nm$^2$ |
|---|---|---|---|---|---|---|
| | $d_1$, nm | $d_2$, nm | | | | |
| PPSSO | 1.280 | 0.4600 | 8.70 | 0.13417 | 2.76 | 0.8368 |
| 1.1 | 1.140 | 0.4200 | 9.10 | 0.27875 | 2.70 | 1.0117 |

Compared to the initial PPSSO, the interplanar spacing decreases in the fractions under consideration. The decrease in the $d_1$ value upon introducing titanium into the siloxane chain is explained by the formation of a coordination bond between the oxygen atoms of one siloxane chain and the vacant d-orbitals of the titanium atom of the neighboring chain. This is also confirmed because the values of the interplanar spacing increase in polymers with a decrease in the content of titanium atoms. Noteworthy are the $d_2$ values, which are mainly responsible for the intrachain distances. For compound 1.1, this parameter decreases, which shows the entry of titanium atoms into the interchain space. These conclusions do not contradict IR spectroscopy, which shows the octahedral environment of the titanium atom.

As for the regions of coherent scattering, there is no dependence on the content of titanium atoms in the polymer. Compared to the initial PPSSO in polymer 1.1, there is a slight decrease in $D_{CSR}$, which can only show a slight increase in internal stresses in crystallites. The data from the electron microscopy of the images (Figure 1) confirm this.

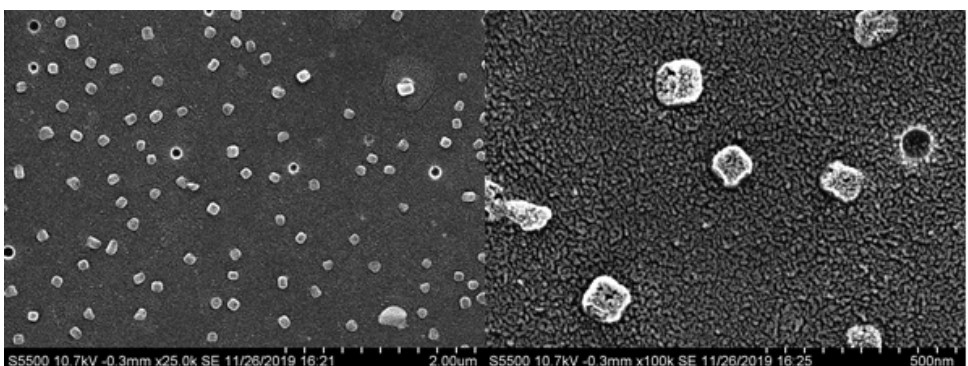

**Figure 1.** Scanning electron microscopy of sample 1.1.

According to scanning electron microscopy, the polymer is packed tightly. Sample 1.1 contains relatively large agglomerates, as well as macropores. The colloidal spheres characteristic of the initial PPSSO [61,62] are absent in the compound under study.

The average diameter of macromolecules ($l$) for compound 1.1 significantly increases as compared to the initial PPSSO (more than twofold), showing not only the incorporation of a titanium atom into the interchain space but also the presence of rather bulky acetylacetonate groups at the ends of the polymer chain.

According to the data from the elemental and X-ray phase analyses and IR spectroscopy, the insoluble fraction is compound 1.2, which has the following composition:

$$[(L^1{}_2TiCl_2)_5(TiO_2)_{0.28}]\cdot[(PhSiO_{1.5})(SiO_2)_{0.18}]$$

The appearance of an absorption band in the IR spectrum at 1134 cm$^{-1}$ (see Figures S2 and S3 in Supplementary Materials) provides confirmation of the presence of PPSSO in compound 1.2 that did not enter the reaction.

In contrast to the results described in [30], the polymerization of the initial titanium complex under conditions of mechanochemical activation did not occur. However, the yield of polytitansiloxane was only 31.6%.

### 3.2. Synthesis Based on Dichloro-bis-(3-oxo-1-phenylbut-1-en-1-olate) Titanium (IV)

To study the influence of the size of the organic ligand at the titanium atom on the ability to cleave the siloxane bond, we used the benzoylacetonate ligand in synthesis 2. When isolating the products of synthesis 2 based on PPSSO and complex 2, three fractions were obtained: 2.1 (yellow), 2.2 (orange), and 2.3 (red). Mass fractions of fractions and the elemental analysis are shown in Table 3.

**Table 3.** Data from elemental analysis of products of synthesis 2.

| Fraction | W, % | Found/Calculated, % | | | | | Yield, % | |
|---|---|---|---|---|---|---|---|---|
| | | Ti | Si | C | Cl | Si/Ti | Ti | Si |
| 2.1 | 21.17 | $(PhSiO_{1.5})_{41}(PhSiO(OH))_{13}(TiO_2)_6(TiOCl_2)_4\ 5L^2H$ | | | | | | |
| | | 5.0/5.3 | 16.9/17.0 | 48.3/50.4 | 3.1/3.1 | 5.7 | 12.43 | 71.82 |
| 2.2 | 41.82 | $[(PhSiO_{1.5})_2(OTiL^2{}_2)]_n \cdot 4L^2{}_2TiCl_2 \cdot 3.5L^2H$ | | | | | | |
| | | 7.7/7.9 | 1.9/1.9 | 58.8/58.3 | 9.4/9.4 | 1:2.4 | 37.80 | 17.62 |
| 2.3 | 37.01 | $10.8(L^2{}_2TiCl_2) \cdot SiO_2$ | | | | | | |
| | | 10.7/10.7 | 0.6/0.6 | 53.4/53.7 | 15.2/15.9 | 1:10.8 | 46.56 | 4.46 |

where $L^2$—

According to the elemental analysis data, the obtained Si/Ti ratio differs from the specified one, and it is 5.7:1 for the high-molecular-weight fraction 2.1. In the IR spectrum (Figure S4 in Supplementary Materials), there is a broad absorption band in the region of 1000–1030 cm$^{-1}$ corresponding to vibrations of the siloxane bond and merging into a doublet with an intense absorption band of the Si-C bond (1132 cm$^{-1}$). The presence of the Ti-O-C bond also increases the absorption in the 1000–1130 cm$^{-1}$ region. Vibrations of the Ti-O- bond (in contrast to the analogous product of the previous synthesis) cannot be unambiguously identified by the singlet signal in the 1260 cm$^{-1}$ region, as it is overlapped by the broadened absorption band of the phenylsiloxane bond and remains in the spectrum as a small shoulder. Low-intensity vibrations of the bonds of benzoylacetonate groups appear in the range of 1357 cm$^{-1}$ (-C-O-Ti), 1488, and 1523 cm$^{-1}$ (C=O).

Stretching and bending vibrations of CH bonds in the ligand and phenyl substituent are manifested in the regions of 1431, 3074, 3051, and 3008 cm$^{-1}$; vibrations of the C=C bond in the acetylacetonate fragment are overlapped by the same bonds of the phenyl substituent and are manifested in the 1595 cm$^{-1}$ region. Noteworthy is the presence of an absorption band in the region of 3631 cm$^{-1}$, which corresponds to the vibrations of free silanol groups [63]. The vibrations of silanol groups can also be identified by the presence of an absorption band at 852 cm$^{-1}$. More intense doublet vibrations of the Si-O bond (497–447 cm$^{-1}$) overlap vibrations of the Ti-O bond, which appear as a narrow singlet in the region of 472 cm$^{-1}$. As in the IR spectrum of the previous synthesis, there are no absorption bands in the region of 540 cm$^{-1}$ (C=O → Ti).

In the $^1$H NMR spectrum, a signal of low intensity is observed at 6.24 ppm, which corresponds to the proton of the b-diketonate ring in the γ-position, as well as a signal tripled in intensity in the region of 2.26 ppm, corresponding to the chemical shifts of the protons of the methyl group. A broadened multiplet in the range 7.1–7.95 ppm corresponds to chemical shifts of protons in the aromatic ring of both the siloxane and the complex.

The molecular weight of the obtained compound is over 7000 (the limit of the separability of the chromatographic column). Whereas the data from the gel permeation chromatography and the NMR spectroscopy did not allow us to determine the molecular weight of the polymer with sufficient approximation, we acylated product 2.1 and titrated the resulting water by the Fisher method. The mass fraction of hydroxyl (silanol) groups

was 2.5%. Calculation of the molecular weight calculated based on the content of hydroxyl groups showed that compound 2.1 has a molecular weight of approximately 8900.

High-molecular compound 2.1 contains the following structural units (Scheme 2):

**Scheme 2.** Structural units of compound 2.1.

General composition of compound 2.1, considering the value of the average molecular weight, corresponds to the following formula:

$$(PhSiO_{1.5})_{41}(PhSiO(OH))_{13}(TiO_2)_6(TiOCl_2)_4 \ 5L^2H.$$

While heating compound 2.1 within 20 min at 200 °C, the absolute weight of the polymer decreased by 5.0%. This indicates insignificant polymerization processes associated with the presence of silanol groups in the polymer. Further heating did not lead to a loss of absolute mass.

Thus, as a result of mechanochemical activation, not only hydrolysis of the initial complex occurred due to the content of associated water in PPSSO, but also the abstraction of the ligand. The high chlorine content is due to the presence of surviving fragments of the titanium complex in the chain.

According to the X-ray phase analysis data (Table 4, Figure S5 in Supplementary Materials), fraction 2.1 is amorphous.

**Table 4.** Data from X-ray phase analysis of PPSSO and fraction 2.1.

| Polymer | Diffraction Peak | | $2\theta(d_1)$, Degree | l, nm | $D_{CSR}$, nm | S, nm$^2$ |
| | $d_1$, nm | $d_2$, nm | | | | |
|---|---|---|---|---|---|---|
| PPSSO | 1.280 | 0.4600 | 8.70 | 0.13417 | 2.76 | 0.8368 |
| 2.1 | 1.239 | 0.4657 | 7.12 | 0.19797 | 4.04 | 0.8826 |

Compared to the initial PPSSO, the interplanar spacing decreases in the fraction under consideration. The decrease in the $d_1$ value upon introducing titanium into the siloxane chain is explained by the formation of a coordination bond between the oxygen atoms of one siloxane chain and the vacant d-orbitals of the titanium atom of the neighboring chain. This is also confirmed because the values of the interplanar spacing increase in polymers with a decrease in the content of titanium atoms. Noteworthy are the $d_2$ values, which are mainly responsible for the intrachain distances. In compound 2.1, this value increases in comparison to the initial PPSSO, which can be explained by incorporating a titanium atom into the polymer chain. These conclusions do not contradict IR spectroscopy, which shows the octahedral environment of the titanium atom.

As for the regions of coherent scattering, there is no dependence on the content of titanium atoms in the polymer. Compared to the initial PPSSO in polymer 2.1, these values

increase rather strongly, which shows the improvement of metastable lamellas. This is confirmed by the data from the electron microscopy of the images (Figure 2).

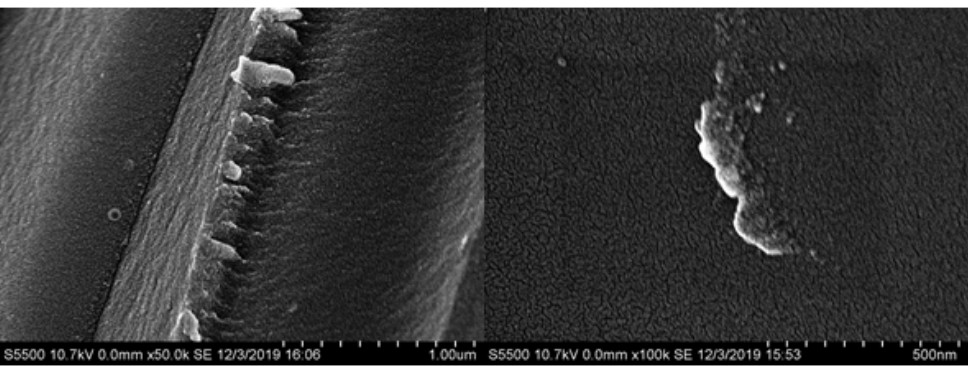

**Figure 2.** Scanning electron microscopy of sample 2.1.

According to scanning electron microscopy, the polymers are packed tightly. Sample 2.1 lacks macropores and agglomerates. In polymer 2.1, relatively regular and dense aggregates are formed, and the formation of a layered structure can be observed. There are no colloidal spheres characteristic of the initial PPSSO in fraction 2.1. We connect their destruction and fusion because of mechanochemical activation.

For compound 2.1, the average macromolecule diameter increases in comparison to PPSSO by 47%. This may also show the partial binding of two neighboring silsesquioxane chains through the titanium atom.

According to the data from the elemental analysis (Table 3), the IR and NMR spectroscopy (Figures S6 and S7 in Supplementary Materials), and the XRD (Figure 3), fraction 2.2 is a mixture comprising polymeric titansiloxane, a coordination polymer, and an initial ligand. The following is a general formula of the mixture: $[(PhSiO_{1.5})_2(OTiL^2_2)]_n \cdot 4L^2_2TiCl_2 \cdot 3.5L^2H$. During gel chromatographic separation of this fraction, the following compounds were isolated separately: polymer titansiloxane $[(PhSiO_{1.5})_2(OTiL^2_2)]_n$ (M > 6000, found/calculated, %: Ti 7.4/7.4, Si 8.6/8.7, C 57.9/59.5) and coordination oligomer $[L^2_2TiCl_2]_4$ (M ≈ 1800, found/calculated, %: Ti 10.7/10.8, Si 0.0/0.0, C 54.7/54.4, Cl 16.0/16.1). Benzoylacetone was not isolated or analyzed separately. The isolated tetramer of the starting complex was destroyed by heating in ethyl alcohol to a monomer.

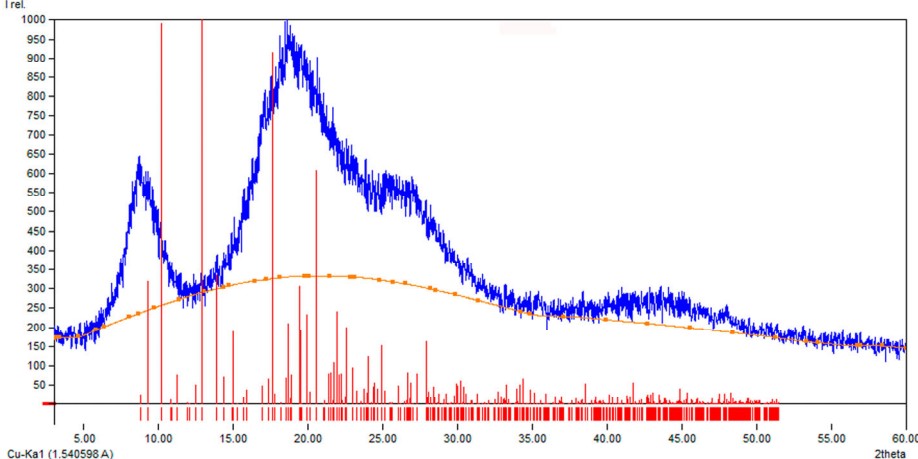

**Figure 3.** Diffractograms of fraction 2.2 (blue) and initial complex (red).

The second high-molecular-weight fraction was observed in a significant amount only for syntheses 2 and 3. Its amorphous state is confirmed by the data from the X-ray phase analysis (Figure 3). In contrast to the previous fraction, for compound 2.2, the appearance

of four halos is observed (Table 5). According to the theory of X-ray diffraction analysis ($\sin^2\theta_{002} = 4\sin^2\theta_{001}$, $\sin^2\theta_{003} = 9\sin^2\theta_{001}$, $\sin^2\theta_{004} = 16\sin^2\theta_{001}$), amorphous halos of the second, third, and fourth orders should be in the angular ranges of diffraction angles $2\theta = 22.34°$, $33.53°$, and $44.72°$. Thus, the diffraction pattern of compound 2.2 exhibits amorphous halos of the second, third, and fourth orders.

**Table 5.** Data from X-ray phase analysis of PPSSO and fraction 2.2.

| Polymer | Diffraction Peak | | | | | | | | $D_{CSR}$, nm |
|---|---|---|---|---|---|---|---|---|---|
| | 1 | | 2 | | 3 | | 4 | | |
| | d, nm | 2θ, deg | d, nm | 2θ, deg | d, nm | 2θ, deg | d, nm | 2θ, deg | |
| PPSSO | 1.280 | 8.70 | 0.460 | 19.18 | - | - | - | - | 2.76 |
| 2.2 | 0.7907 | 11.18 | 0.461 | 19.23 | 0.342 | 26.00 | 0.206 | 43.90 | 1.76 |

Given the additional halos, it can be noted that the presented fractions are not only in an amorphous state, but that they also have a certain ordering.

It is inappropriate to carry out a comparative analysis of the X-ray diffraction patterns of the obtained compounds with PPSSO, as the fractions are mixtures of polymeric titanium phenylsiloxanes and polymerized coordination complexes. However, the analysis of diffraction patterns showed the presence in the mixture, besides the polymer product, of initial complexes in oligomeric states, which confirms the corresponding conclusions made earlier. The amorphization of the initial complexes, as well as the broadening of the diffraction maxima, are primarily associated with microdistortions in crystals, a decrease in crystallites, and the formation of oligomeric products and agglomerates. In Figure 3, the diffraction maxima obtained from the experiment are highlighted in blue, and the diffraction patterns of the initial titanium complexes are highlighted in red.

These conclusions are also confirmed by the electron microscopy data. Figure 4 shows that the fraction comprises agglomerates of a porous structure of various shapes and sizes. Fraction 3.2 has a more ordered structure, which we can observe both on the X-ray (Figure 3) and on the micrograph (Figure 4).

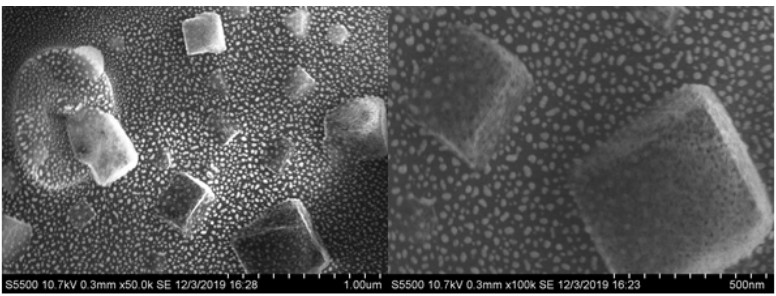

**Figure 4.** Scanning electron microscopy of sample 2.2.

The presented data from the electron microscopy and the X-ray phase analysis correlate well with each other. In addition, the size of the coherent scattering region for fraction 2.2 is more than 50% lower than that of the initial PPSSO.

According to the data from the elemental and X-ray phase analyses (Figure S8 and Table S1 in Supplementary Materials) and IR spectroscopy, the insoluble fraction (2.3) is the initial titanium complex with a small content of silicon oxide: $10.8(L^2_2TiCl_2)\cdot SiO_2$.

### 3.3. Synthesis Based on Dichloro-bis-(3-oxo-1,3-diphenylprop-1-en-1-olate) Titanium (IV)

In synthesis 3, we used a titanium complex with a more sterically hindered ligand, dibenzoylmethanate.

After mechanochemical activation, three fractions were isolated: a yellow high-molecular-weight fraction (3.1), an orange-red high-molecular-weight fraction (3.2), and

a red low-molecular-weight fraction (3.3). Elemental analysis of the fractions is shown in Table 6.

**Table 6.** Data from elemental analysis of products of synthesis 3.

| Fraction | W, % | Found/Calculated, % | | | | | Yield, % | |
|---|---|---|---|---|---|---|---|---|
| | | Ti | Si | C | Cl | Si/Ti | Ti | Si |
| 3.1 | 22.77 | $[(PhSiO_{1.5})_{5.2}(PhSiO(OH))_{1.1}(TiO_2)_{0.9}(OTiL^3{}_2)_{0.1}\,0.3L^3H]_n$ | | | | | | |
| | | 4.7/4.7 | 17.3/17.3 | 52.8/53.5 | - | 6.3 | 11.80 | 74.14 |
| 3.2 | 53.90 | $[(PhSiO_{1.5})(OTiL^3{}_2)_{0.9}(TiO_2)]_n \cdot 1.9(L^3{}_2TiCl_2)$ | | | | | | |
| | | 10.4/10.4 | 1.6/1.6 | 62.2/62.0 | 7.9/7.7 | 1:3.8 | 61.79 | 16.23 |
| 3.3 | 23.33 | $L^3{}_2TiCl_2\,0.06SiO_2\,0.9H_2O$ | | | | | | |
| | | 8.2/8.2 | 0.3/0.3 | 60.0/61.4 | 12.2/12.1 | 1:16 | 21.10 | 1.32 |

where $L^3$—

According to the elemental analysis data, the obtained Si/Ti ratio differs from the specified one and is equal to 6.3. In contrast to the previous synthesis, there is no chlorine in the analogous high-molecular-weight fraction 3.1. In addition, according to the IR spectroscopy data (Figure S9 in Supplementary Materials), there are a small number of diketonate groups, both free and bound to titanium (1554 and 1521 cm$^{-1}$).

In addition, in the IR spectrum, vibrations of bonds of hydroxyl groups (3631 and 854 cm$^{-1}$) and associated water (3406 cm$^{-1}$) are observed; there is an intense absorption band in the region of 1028 cm$^{-1}$ characteristic of antisymmetric vibrational vibrations of the siloxane bond. Signals in the range of 3008, 3051, 3072, 2920, 1597, 1431, 1132, and 696 cm$^{-1}$ correspond to bond vibrations in phenyl substituents and Si-Ph (Si-C, C=C, C-H). The Ti-L bond can be identified by the vibration in the 1355 cm$^{-1}$ region and the Si-O-Ti bond at 920 cm$^{-1}$, which is overlapped by an intense absorption band of the siloxane bond and is observed in the spectrum as a small shoulder.

The average molecular weight of compound 3.1, according to the gel permeation chromatography data, exceeds the column divisibility limit and is more than 6000. Heating to 200 °C did not lead to a noticeable weight loss (less than 2%), which allows us to conclude that condensation processes are insignificant, including the absence of chlorine atoms in the polymer.

The NMR spectrum on the proton nuclei confirms the insignificant content of diketonate groups (a weak and low-intensity signal in the region of 6.43 ppm, corresponding to the proton in the γ-position of the diketonate ring).

Based on GPC, elemental analysis, IR, and NMR spectroscopy, it can be concluded that the high molecular weight fraction 3.1 corresponds to the compound of the following general formula: $[(PhSiO_{1.5})_{5.2}(PhSiO(OH))_{1.1}(TiO_2)_{0.9}(OTiL^3{}_2)_{0.1}\,0.3L^3H]_n$. Structurally, the elements of compound 3.1 can be depicted as follows (Scheme 3):

**Scheme 3.** Structural units of compound 3.1.

According to X-ray phase analysis data (Table 4, Figure S10 in Supplementary Materials), fraction 3.1 is amorphous.

Compared to the initial PPSSO, the interplanar spacing decreases in the fractions under consideration (Table 7). The decrease in the $d_1$ value upon introducing titanium into the siloxane chain is explained by the formation of a coordination bond between the oxygen atoms of one siloxane chain and the vacant d-orbitals of the titanium atom of the neighboring chain. This is also confirmed because the values of the interplanar spacing increase in polymers with a decrease in the content of titanium atoms in it. Noteworthy are the $d_2$ values, which are mainly responsible for the intrachain distances. In compound 3.1, this value increases in comparison to the initial PPSSO, which can be explained by incorporating a titanium atom into the polymer chain. These conclusions do not contradict IR spectroscopy, which shows the octahedral environment of the titanium atom.

**Table 7.** Data from X-ray phase analysis of PPSSO and fraction 3.1.

| Polymer | Diffraction Peak | | $2\theta(d_1)$, Degree | l, nm | $D_{CSR}$, nm | S, nm$^2$ |
|---------|--------|--------|--------|--------|--------|--------|
| | $d_1$, nm | $d_2$, nm | | | | |
| PPSSO | 1.280 | 0.4600 | 8.70 | 0.13417 | 2.76 | 0.8368 |
| 3.1 | 1.244 | 0.4622 | 7.10 | 0.12202 | 3.87 | 0.8768 |

Compared to the initial PPSSO, in polymer 3.1, there is a slight increase in $D_{CSR}$, which can only show a slight decrease in internal stresses in crystallites. In the case of polymer 3.1, these values increase rather strongly, which shows the improvement of metastable lamellas. This is confirmed by the data from the electron microscopy of the images (Figure 5).

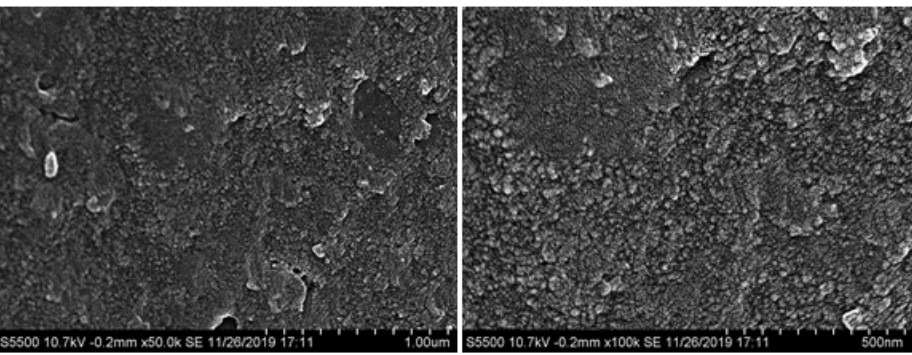

**Figure 5.** Scanning electron microscopy of sample 3.1.

Sample 3.2 lacks macropores and agglomerates. In polymer 2.1, relatively regular and dense aggregates are formed, while in polymer 3.1, the formation of a layered structure can be observed. There are no colloidal spheres characteristic of the initial PPSSO in any of the three studied compounds.

In compound 3.1, the average diameter of the macromolecule becomes smaller because of the lower content of ligands at the titanium atom; this makes possible the "constriction" of polymer chains because of the donor–acceptor interaction.

As in synthesis 2, as a result of mechanochemical activation, fraction 3.2 was isolated; it is a mixture of polytitanphenylsiloxane and the initial complex (Figures S8 and S9 in Supplementary Materials). The following is a general formula of the mixture: $[(PhSiO_{1.5})(OTiL^3_2)_{0.9}(TiO_2)]_n \cdot 1.9(L_2TiCl_2)$. The mixture was separated using GPC. A polymer with the composition $[(PhSiO_{1.5})(OTiL^3_2)_{0.9}(TiO_2)]_n$ and a titanium complex were isolated. However, in contrast to the previous synthesis, the high-molecular-weight compound was not polytitanphenylsiloxane, but rather a mixture comprising polyphenylsiloxane and a polymer titanium complex. This was determined by dissolving this fraction into toluene: polyphenylsilsesquioxane went into the solution, and the polymer titanium complex was

filtered off. Thus, fraction 3.2 was divided into three compounds: $[PhSiO_{1.5} \cdot 0.07H_2O]_n$ (m = 0.2094 g, M > 6000, found/calculated, %: Si 21.5/21.5, C 54.7/55.3); $[(L^3_2TiO)_{0.9}(TiO_2)]_n$ (m = 0.9069 g, M > 6000, found/calculated, %: Ti 17.1/16.9, C 61.1/60.1); and $L^3_2TiCl_2$ (m = 1.7237 g, M = 565, found/calculated, %: Ti 8.1/8.5, C 63.6/63.7, Cl 13.0/12.6).

In the IR spectrum of the polymerized titanium complex, vibrations of the bonds of the phenyl substituents of the dibenzoylacetonate ligand (2862, 2925, 2974, 1593 cm$^{-1}$), the C=C and C=O bonds of the diketone (1517 and 1487 cm$^{-1}$), are observed. The vibrations of the Ti-O bond bound to the ligand are manifested near 1350 cm$^{-1}$, and the titanoxane bond (Ti–O–Ti) is manifested near 771 cm$^{-1}$ because of the symmetric stretching vibrations of the Ti–O bonds of the TiO$_4$ tetrahedron [64]. X-ray phase analysis confirms the polymeric character of this fraction. It is assumed that the polymerized titanium complex has the following cyclolinear structure (Scheme 4):

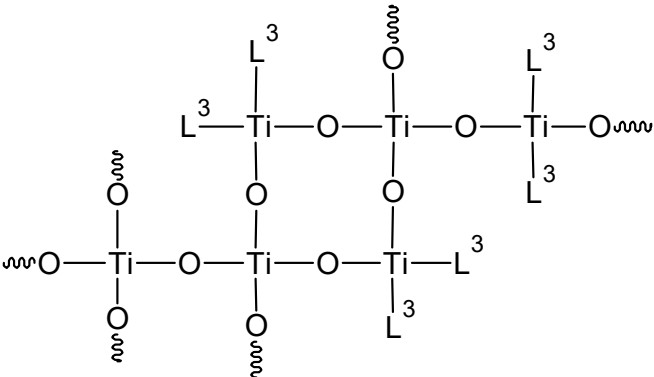

**Scheme 4.** Structural units of compound 3.2.

The second high-molecular-weight fraction is observed in a significant amount only for syntheses 2 and 3. Its amorphous state is confirmed by the data from the X-ray phase analysis (Figure 6). In contrast to the previous fractions, for compound 3.2, the appearance of four halos is observed (Table 8). Thus, the diffraction pattern of compound 3.2 exhibits amorphous halos of the second, third, and fourth orders. Given the additional halos, it can be noted that the presented fractions are not only in an amorphous state, but that they also have a certain ordering.

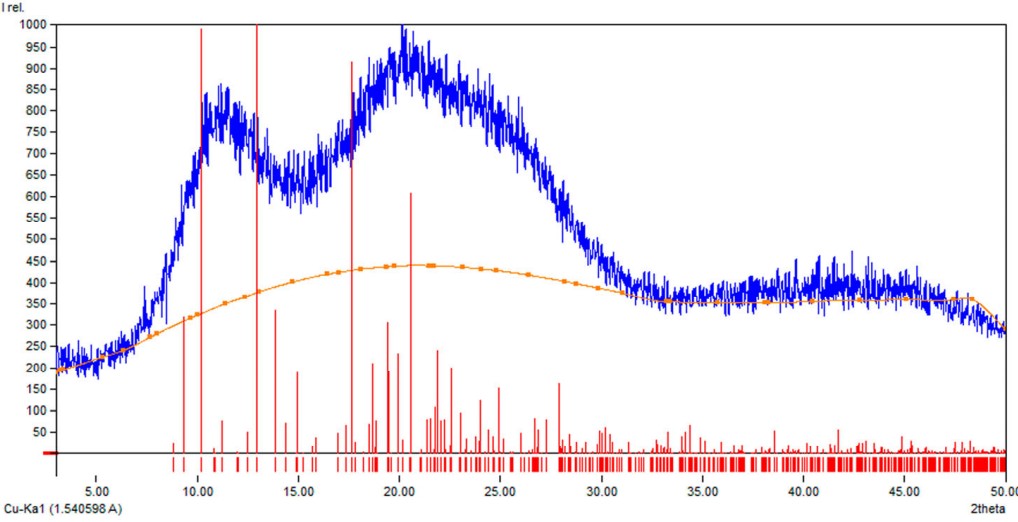

**Figure 6.** Diffractogram of fraction 3.2 (blue) and initial complex (red).

**Table 8.** Data from X-ray phase analysis of PPSSO and fraction 3.2.

| Polymer | Diffraction Peak | | | | | | | | $D_{CSR}$, nm |
|---|---|---|---|---|---|---|---|---|---|
| | 1 | | 2 | | 3 | | 4 | | |
| | d, nm | 2θ, deg | d, nm | 2θ, deg | d, nm | 2θ, deg | d, nm | 2θ, deg | |
| PPSSO | 1.280 | 8.70 | 0.460 | 19.18 | - | - | - | - | 2.76 |
| 3.2 | 1.0200 | 8.66 | 0.479 | 18.50 | 0.343 | 25.95 | 0.205 | 44.17 | 3.29 |

It is inappropriate to carry out a comparative analysis of the X-ray diffraction patterns of the obtained compounds with PPSSO, as the fractions are mixtures of polymeric titanium phenylsiloxanes and polymerized coordination complexes. However, the analysis of diffraction patterns showed the presence in the mixture, besides the polymer product, of initial complexes in oligomeric states, which confirms the corresponding conclusions. The amorphization of the initial complexes, as well as the broadening of the diffraction maxima, are primarily associated with microdistortions in crystals, a decrease in crystallites, and the formation of oligomeric products and agglomerates. In Figure 6, the diffraction maxima obtained from the experiment are highlighted in blue, and the diffraction patterns of the initial titanium complexes are highlighted in red.

These conclusions are also confirmed by the data from the electron microscopy. Figure 7 shows that as a result of mechanochemical activation, a fraction is formed, the morphology of which comprises agglomerates of a porous structure of various shapes and sizes. Fraction 3.2 has a more ordered structure, which we can observe both on the X-ray and on the micrograph. As seen from the results of scanning electron microscopy, there are no agglomerates, spherical formations, or colloidal spheres capable of further coalescence in fraction 3.2. The structure of the joint is uniform, but there are many micro-cracks.

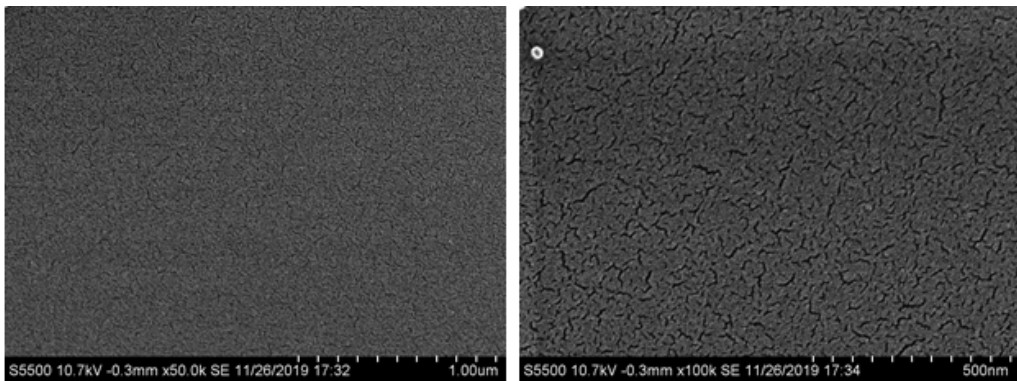

**Figure 7.** Scanning electron microscopy of sample 3.2.

The presented data from electron microscopy and X-ray phase analysis correlate well with each other. In addition, regarding the size of the coherent scattering region for fraction 3.2, the opposite is observed: the CSR size increases with increasing ordering in the structure.

According to the data from the elemental and X-ray phase (Figure S13 and Table S2 in Supplementary Materials) analyses and IR and NMR spectroscopy, the insoluble fraction (3.3) is the initial titanium complex with an insignificant content of hydrated silicon oxide: $L^3_2TiCl_2$ $0.06SiO_2$ $0.9H_2O$.

## 4. Conclusions

In the presented work, the possibility of synthesizing polytitanphenylsiloxanes under conditions of mechanochemical activation was shown. For the first time, the mechanochemical synthesis of polytitanphenylsiloxanes based on PPSSO and various titanium b-diketonate complexes has been carried out. It has been established that high-molecular-weight products are formed with a Si/Ti ratio differing from the ratios. The resulting ratios increase with increasing size of the ligand at the titanium atom. With an increase in the volume of the organic ligand, titanium atoms enter the polymer siloxane chain to a lesser extent. It has been proven that under the conditions of mechanochemical activation with the benzoylacetonate complex, an oligomeric product is formed, which is a coordination tetramer. When titanium dichloride dibenzoylmethanate is used, a significant part of it decomposes to form its own polymer product.

The characteristics of the remaining organic substituents of silicon and titanium atoms can be used for the synthesis of homogeneous materials. This is achieved by thermal treatment of the resulting macromolecular compounds. The relatively high resistance to ultraviolet radiation and the high hydrophobicity of the obtained polymers are used to create protective coatings. High-temperature processing of polytitanphenylsiloxanes produces titanium silicate nanocomposites used as oxidation catalysts, which can later replace platinum and palladium catalysts used in the automotive industry.

**Supplementary Materials:** The following supporting information can be downloaded at: https://www.mdpi.com/article/10.3390/powders2020027/s1.

**Author Contributions:** Conceptualization, V.L. and A.K.; methodology, V.L., A.T. and A.R.; validation, V.L., A.K. and N.S.; formal analysis, V.L., A.T. and A.R.; investigation, V.L., A.K., A.T. and A.R.; resources, V.L.; data curation, V.L., A.K. and N.S.; writing—original draft preparation, V.L., A.T. and A.K.; writing—review and editing, V.L., A.K. and N.S.; project administration, A.K. and N.S. All authors have read and agreed to the published version of the manuscript.

**Funding:** This research received no external funding.

**Institutional Review Board Statement:** Not applicable.

**Informed Consent Statement:** Not applicable.

**Data Availability Statement:** The data presented in this study are available on request from the corresponding author. The data are not publicly available because they are part of an ongoing study.

**Acknowledgments:** The authors are grateful to Karabtsov A.A. (Laboratory of X-ray Methods, Far East Geological Institute, Far Eastern Branch Russian Academy of Sciences) and Kuryavoy V.G. (Institute of Chemistry, Far Eastern Branch of the Russian Academy of Sciences) for carrying out some physicochemical research methods.

**Conflicts of Interest:** The authors declare no conflict of interest.

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
