# Peer review of "Interaction of Polyphenylsilsesquioxane with Various β-Diketonate Complexes of Titanium by Mechanochemical Activation"

_2674-0516, doi:10.3390/powders2020027_

Round 1
Reviewer 1 Report
Language can be improved further
NMR interpretation should be more precise.
Please provide expanded NMR spectrum for individual complex
Provide latest details about Zeigler Nata catalysis
Author Response
We express our gratitude for the attention paid to our work. Your comments and suggestions have been included in the text of the article.

Reviewer 2 Report
The contribution is interesting and has many experimental results. Perhaps the way they are presented is not the more explicit but the contents and analysis are worth publication. Please attend to the following details in order to enhance the quality and clearness of this piece of work.
Please include at least two references from this journal “Powders” related to your work.
English must be revised accordingly in the entire contribution, as well as to provide grammatical enhancement in many phrases like those commonly used by the authors, e.g. “Someone widely …”.
In the abstract and during the contribution please always use “high-molecular-weight”.
It is also highly recommended the employment of third person writing during the entire contribution, e.g in the abstract, the following sentence is quite pedantic as well as fuzzy “I aim the work at finding effective and environmentally friendly methods for the synthesis and modification of element organic high-molecular compounds.”, please rephrase as requested.
Another recommendation in the abstract is that if the work provided the stated in this paragraph “The work is fundamental and contributes to the understanding of the mechanisms of mechanochemical synthesis.”, please better include your findings instead of this cloudy sentence.
In the phrase in lines 35-36 please include at least two references for these findings.
As has been already underlined, there are several errors, some will be included herein but a major revision is required, like this in lines 99-100 “Synthesis of Polyphenylsilsesquioxane. Polyphenylsilsesquioxane (PPSSO) wassynthesized as we described previously it [14].”
Please use “mL” instead of “ml” in general.
In the experimental section please specify the method that you employed to determine the yields of the final products. And in other cases you are not reporting the yield, please include these missing values.
It has been considered to include a general reactions diagram for a more chemical comprehension of the studied reactions, and not only cite them with references [12,14] and undesirable solvent side reactions [25].
In Table 1 you are including L₂ which has not been defined nor mentioned anywhere, they should be the acetonates indicated in the experimental section, please define this ligand in due course, as well as L¹. Additionally, use super, or subscript for them, but the same nomenclature for both.
The IR analysis has been thoroughly determined, nevertheless, there are some minor details that need amendment. In lines 230-232 you state that “Noteworthy is the absence of an absorption band near 540 cm⁻¹, which shows the π-bonding of the carbonyl group of the acetylacetonate fragment with the titanium atom (C=O→Ti) [52].”, please revise this since the coordinative bonding between C=O and Ti is not of π nature at all. And if this band is not appearing at that frequency therefore another type of bonding is occurring, please determine which band corresponds to this structural moiety. Some clue is present at the ¹H NMR since the acetylacetonate signals appear as asymmetric, please further revise in order to understand the bonding scheme.
In line 234 please use superscript for the NMR nuclei “According to the data of ¹H NMR spectroscopy”.
According to the structure that you present in line 251, please determine the coordination number to Ti where two L¹ ligands are present. Also herein, please specify if the initial Ti-acetylacetonate is serving as the polymer initiator?? Or what should be the polymerization mechanism herein?? Also consider drawing your structural representation in a more convincing 3D model, this according to what you state in lines 276-285.
Other grammatical errors have been found, one in line 259 “It known [54] that two diffraction maxima appear”, please correct this, and another in line 289 “The data of electron microscopy of the images (Fig. 1 confirms this).”.
Please be more explicit with this sentence in lines 295-296 “We connect with their destruction and fusion because of mechanochemical activation.”.
In lines 314-315 please correct this paragraph “In contrast to the previous synthesis, three fractions was isolated after”
What happened to the acetonate ligand in compound 2.1, it is not present in the polymeric structure, and therefore is occluded within the polymer, what findings are you leading to state this??
For the tables of X-ray phase analysis, please reconsider the use of “reflex”.
In all the figures and tables, please end with “.” final point.
Author Response

(The authors gave the same response as above.)
